# Preformulation Studies of Novel Menthol Prodrugs with Antiparasitic Activity: Chemical Stability, In Silico, and In Vitro Permeability Assays

Camila M. Clemente [1,2,*] , Renée Onnainty [3] , Nadina Usseglio [3] , Gladys E. Granero [3] and Soledad Ravetti [4,*]

1   Instituto Académico Pedagógico de Ciencias Básicas y Aplicadas, Universidad Nacional de Villa María, Villa María 5900, Argentina
2   Departamento de Química Biológica, Facultad de Ciencias Exactas y Naturales, Universidad de Buenos Aires (FCEyN-UBA) e Instituto de Química Biológica de la Facultad de Ciencias Exactas y Naturales (IQUIBICEN) CONICET, Pabellón 2 de Ciudad Universitaria, Buenos Aires C1428EHA, Argentina
3   Unidad de Investigación y Desarrollo en Tecnología Farmacéutica (UNITEFA-CONICET), Departamento de Ciencias Farmacéuticas, Facultad de Ciencias Químicas, Universidad Nacional de Córdoba, Córdoba 5000, Argentina; ronnainty@unc.edu.ar (R.O.); nusseglio@unc.edu.ar (N.U.); glagranero@unc.edu.ar (G.E.G.)
4   Centro de Investigaciones y Transferencia de Villa María (CIT VM), Instituto Académico Pedagógico de Ciencias Humanas, Villa María 5900, Argentina
*   Correspondence: camilamaraclemente@gmail.com (C.M.C.); sravetti@unvm.edu.ar (S.R.)

**Abstract:** Based on the demonstrated and reported trypanocidal, leishmanicidal, and antiplasmodial activities of two menthol prodrugs, it was decided to proceed with preformulation studies, which are of key relevance in the drug discovery process. The aim of this study is to examine the stability and permeability of two new menthol prodrugs with antiparasitic activity. To determine the stability of menthol and its prodrugs, the corresponding studies were carried out in buffered solutions at pH values of 1.2, 5.8, and 7.4 at 37 °C. In silico permeability studies were performed using the free PerMM software and then in vitro permeability studies were performed using a biomimetic artificial membrane (BAM). Permeability studies conducted in silico predicted that both menthol and its prodrugs would pass through biological membranes via flip-flop movement. This prediction was subsequently confirmed by in vitro BAM permeability studies, where it was observed that the menthol prodrugs (**1c** and **1g**) exhibited the highest $P_{app}$ (apparent permeability) value compared to the parent compound. The study reveals that menthol prodrugs exhibit stability at a pH of 5.8 and possess sufficient in vitro permeability values as preformulation parameters.

**Keywords:** preformulation; antiparasitic; permeability; menthol; prodrugs

## 1. Introduction

The primary constituents of essential oils, commonly referred to as their main molecules, have garnered significant attention in the quest for novel chemical entities to be utilized in drug development [1–4]. Essential oils are complex mixtures of various bioactive compounds, predominantly consisting of terpenoids, phenolic compounds, and other volatile organic compounds. These natural constituents have demonstrated diverse biological activities, such as antimicrobial [5–8], anti-inflammatory [9–11], antioxidant [4,12–14], and anticancer properties [15–17], among others. Incorporating essential oil main molecules into drug development programs demands a comprehensive understanding of their physicochemical properties, pharmacokinetics, and safety profiles. Rigorous preclinical evaluations are essential to ascertain their efficacy, safety, and potential interactions with other drugs.

According to Bakkali et al. [18], approximately 90% of the compounds in essential oils are classified as monoterpenes, among which notable examples are menthol, thymol, and eugenol. This growing interest in essential oil constituents stems from their potential

therapeutic properties and their ability to offer promising leads for drug discovery and development. The prevalence of monoterpenes in essential oils highlights their significance in the pursuit of novel and effective medications. As such, researchers are increasingly exploring the pharmacological properties and applications of these compounds to harness their potential in the field of medicine.

The monoterpene menthol, chemically characterized as 2-isopropyl-5-methylcyclohexane, is a key constituent in the essential oils of various Mentha species. Numerous in vitro and in vivo investigations have documented diverse biological attributes associated with this compound. These include analgesic, antibacterial, antiparasitic, antifungal, anesthetic, and penetration-enhancing activity, as well as chemopreventive and immunomodulatory actions [19–28].

In spite of the various biological activities ascribed to menthol, its effectiveness in medical applications is hindered by its limited physicochemical stability and low bioavailability. [29]. The aforementioned factors motivated us to design different menthol prodrugs in order to improve its pharmacokinetic and pharmacodynamic properties. The synthetic strategy used was the association of menthol with different aliphatic alcohols, these being the molecular complement of the prodrug. Consequently, we carried out the design, synthesis, in silico, in vitro, and, in some cases, in vivo assays, of the antiparasitic activity of these prodrugs against *Trypanosoma cruzi*, *Leishmania braziliensis*, *Plasmodium falciparum*, and *Echinococcus multilocularis* [21–24].

Given the enhanced antiparasitic characteristics demonstrated by these derivatives in comparison to the original compound, these novel variations present themselves as highly encouraging contenders for the treatment of parasitic diseases. The intrinsic stability of these prodrugs emerges as a pivotal factor in elucidating their efficacy. It is worth emphasizing that the mechanisms of action underlying these newly developed compounds rely on the chemical/enzymatic hydrolysis of the -OH bonds situated between menthol and its corresponding molecular complement. Furthermore, in the realm of drug discovery and the intricate design and formulation of pharmaceutical and biopharmaceutical products, permeability stands as another critical parameter warranting meticulous consideration.

Considering the potential of these new menthol prodrugs as antiparasitic candidates, one of the main aims of the work was to determine the stability of compounds **1c** and **1g** (Figure 1) in aqueous solutions at different pH values of biological relevance, as well as the development and validation of gas chromatographic (GC) method to quantify menthol. We also described the permeability of the prodrugs and their parent compound by in silico and in vitro assays using a biomimetic artificial membrane (BAM) simulating the behavior of eukaryotic cell membranes.

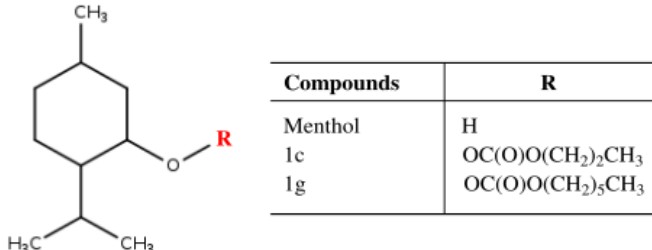

| Compounds | R |
|---|---|
| Menthol | H |
| 1c | $OC(O)O(CH_2)_2CH_3$ |
| 1g | $OC(O)O(CH_2)_5CH_3$ |

**Figure 1.** Menthol and its synthesized prodrugs.

## 2. Results

### 2.1. Stability Studies

2.1.1. Validation of GC Method

The described gas chromatography (GC) method proves to be a straightforward and highly efficient process for the assessment of the stability of menthol and its prodrugs. Table 1 presents key analytical data obtained using this method. The standard curves were thoroughly and comprehensively described through the implementation of linear least squares regression. The calibration data provided in-depth insights into the excellent linearity exhibited over the concentration ranges, and the correlation coefficient (r) value

of 0.99 further attested to the robustness of the results. Notably, no interferences arising from endogenous substances were detected in any of the samples containing menthol or its synthesized derivatives, ensuring the accuracy and reliability of the analysis. Moreover, Table 1 presents the values for the limits of detection (LOD) and quantification (LOQ). Precision was determined by calculating the relative standard deviation (RSD) based on six injections and retention time (tR).

**Table 1.** Statistical significance for menthol calibration curves.

| | | Menthol Validation Parameters | |
|---|---|---|---|
| | | $a \pm (s\ t)$ | $1.8 \times 10^4 \pm (1.8 \times 10^2)$ |
| y = ax + b | | $b \pm (s\ t)$ | $0.07 \pm 0.01$ |
| | | r | 0.99 |
| Limit of detection (LOD) | | | $3.9 \times 10^{-6}$ |
| Limit of quantification (LOQ) | | | $9.4 \times 10^{-3}$ |
| Recovery (% REC) | | | 100.0; 102.7; 100.5 |
| Precision (RSD) | | | 1.4; 3.7 |
| Retention time ($t_R$) | | Menthol | 6.07 |
| | | **1c** | 8.57 |
| | | **1g** | 12.64 |

Recovery was evaluated by calculating the percentage deviation between the expected and observed concentrations. All of these findings collectively highlight the robustness and accuracy of the GC method employed, making it a valuable tool for the study and evaluation of menthol and its prodrugs' stability.

### 2.1.2. Identification of Degradation Compounds

Degradation investigations for **1c** and **1g** were conducted in both acidic and alkaline environments at 70 °C to identify potential degradation products. Across all conditions, the sole detected product in the reaction mixtures was menthol, which is the parent compound for these prodrugs. The identification of compounds **1c**, **1g**, and menthol was confirmed by comparing their retention times with those of standard samples using GC techniques.

When employing GC analysis, no indication of interference between menthol and compounds **1c** and **1g** was detected, underscoring the feasibility of quantifying these prodrugs even in the presence of their degradation product. As illustrated in Figure 2, the GC chromatogram for **1c** (tR = 8.57) exhibits the coexistence of its degradation product menthol (tR = 6.09).

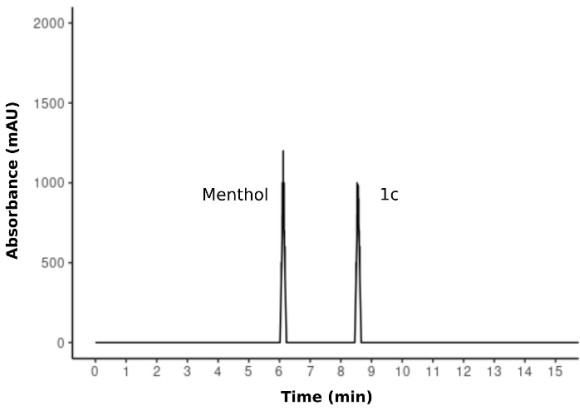

**Figure 2.** Gas Chromatography (GC) for 1c (tR = 8.57) and its parent compound, menthol (tR = 6.09), at a pH of 1.2 and 70 °C.

The observed behavior in these stability studies demonstrated that these compounds exhibit a clear prodrug-like behavior, emphasizing the importance of comprehending the degradation pathways and their implications for compound stability and efficacy.

### 2.1.3. Aqueous Stability

The chemical stability studies of derivatives **1c** and **1g** were conducted in accordance with established experimental protocols, which closely mimic biologically relevant environments. The selected pH values of 1.2, 5.8, and 7.4, along with the physiological temperature of 37 °C, aimed to simulate the conditions of biological relevance. The degradation of all prodrugs under their corresponding pH conditions, assessed by monitoring the prodrug's unaltered concentration over time, consistently adhered to pseudo-first-order kinetics for a duration of more than two half-lives.

In Table 2, the rate constants that describe the conversion of **1c** and **1g** into menthol are shown. These constants were determined using a linear regression analysis that involves plotting the natural logarithm of concentration against time, following the principles of pseudo-first-order kinetics. Additionally, Table 2 includes the half-life values calculated from the hydrolysis rate constants in buffer solutions. Significantly, the compounds displayed discernible variations in stability contingent upon the specific pH conditions within their respective environments.

**Table 2.** Rate constant and half-lives ($t_{1/2}$) of the chemical hydrolysis of **1c** and **1g** at 37.0 °C.

| Compound | pH 1.2 | | pH 5.8 | | pH 7.4 | |
|---|---|---|---|---|---|---|
| | $k$ (min$^{-1}$) | $t_{1/2}$ (min) | $k$ (h$^{-1}$) | $t_{1/2}$ (h) | $k$ (h$^{-1}$) | $t_{1/2}$ (h) |
| **1c** | 0.007 | 99.00 | 0.007 | 99.00 | 0.010 | 69.31 |
| **1g** | 0.007 | 99.00 | 0.007 | 115.52 | 0.010 | 69.31 |

At a pH of 1.2, all the investigated compounds displayed increased susceptibility to degradation, with observable rapid degradation occurring within a matter of minutes. In contrast, in the 5.8 pH environment, the prodrugs demonstrated enhanced stability, exhibiting substantially prolonged $t_{1/2}$ values ranging from 99 h to 115 h. Similar trends were discerned at a pH of 7.4, although the $t_{1/2}$ values were comparatively lower than those observed at pH of 5.8. These findings align with previous literature reports, indicating that carbonate is relatively stable at a pH of 5.8 [23,30,31].

It is noteworthy that these results offer valuable insights into the behavior of the investigated compounds, specifically in terms of their stability and degradation kinetics at different pH levels. Understanding these pH-dependent variations is of utmost importance in designing and formulating pharmaceutical products. From the assays performed, we observed that the hydrolysis of carbonate prodrugs by both acidic and basic catalytic mechanisms yields $CO_2$ and the corresponding alcohols as degradation products. The general hydrolysis reaction of the prodrugs proceeds according to Figure 3.

**Figure 3.** General hydrolysis reaction of menthol prodrugs.

### 2.2. Permeability Studies

2.2.1. In Silico Assays

The estimation of the mean potential force profile (PMF) across the membrane, as depicted in Figure 4, and the determination of the permeability coefficient across a dioleoyl–phosphatidylcholine (DOPC) bilayer membrane, as summarized in Table 3, for the compounds under investigation were conducted through the utilization of the PerMM software (see Section 4 for detailed explanation about software utilization). This computational tool enables the prediction of key physicochemical properties related to the interaction of the compounds with the lipid bilayer. The PMF analysis offers valuable insights into the energy landscape governing the translocation of the compounds across the membrane, while the permeability coefficient data provides quantitative information on their diffusion through the lipid bilayer. Such comprehensive assessments contribute significantly to the understanding of the compounds' behavior at the molecular level and are essential in evaluating their potential applicability and efficiency in various biomedical and pharmaceutical contexts.

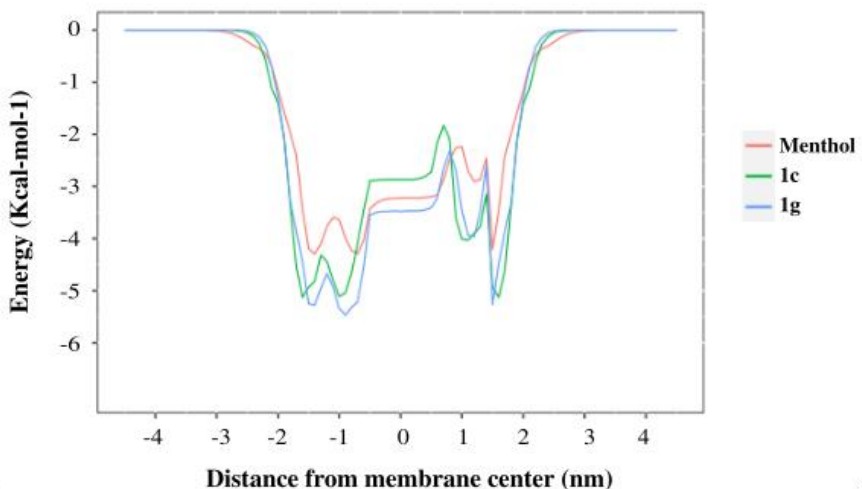

**Figure 4.** Passive permeability predictions calculated with the PerMM server for menthol, **1c**, and **1g**.

**Table 3.** Permeability predictions as calculated using the PerMM server.

| Compounds | Free Energy of Binding (DOPC) |
|---|---|
| Menthol | −4.32 |
| **1c** | −5.13 |
| **1g** | −5.41 |

From the analysis of the results shown in Figure 4, it is observed that all compounds showed negative energy profiles at all depths of the graph with a steady state at the center of the membrane. The studied compounds presented high ClogP values, so, from the analysis, it can be observed that their partitioning will be higher in the hydrophobic inner region of the lipid bilayer as opposed to the lipid–water interface within the membrane.

For all compounds, it is predicted that the energy minimum is found at the moment of contact with the first lipid layer ($\approx$−2 nm), and the higher the lipophilicity of the molecule, the higher the binding free energy it presents. Once the compounds cross the first lipid layer, an energy barrier appears before crossing the second lipid layer. Since the energy profiles are similar to those observed in the compounds described by Mollazadeh et al., it is possible a "flip-flop" movement occurs in these compounds, facilitating quicker "flip-flop" movements between layers and, in theory, attaining higher concentrations on the cytoplasmic side [23]. The permeability coefficients predicted by PerMM strengthen the previous assumptions (Table 3). Whereas, menthol derivatives, being more hydrophobic,

and having higher permeability coefficients, are more predisposed to accumulation in the cytoplasm.

### 2.2.2. In Vitro Assays

The research conducted on apparent permeability ($P_{app}$) and steady-state fluxes (Jss) for menthol and its two derivatives (**1c** and **1g**) is of great significance in understanding the transport mechanisms of these compounds. Table 4 shows the results of $P_{app}$ and Jss for three different compounds: menthol, **1c**, and **1g**. Additionally, the value of the octanol–water partition coefficient (cLogP) is shown for each compound.

**Table 4.** Calculated octanol/water partition coefficient (cLogP), steady-state fluxes (Jss), and $P_{app}$ of menthol, menthol–propanol (**1c**), and menthol–hexanol (**1g**).

| Compound | cLogP | $P_{app}$ ($10^{-4}$) (cm.s$^{-1}$) | Jss ($10^{-5}$) (mg.s$^{-1}$) |
|:---:|:---:|:---:|:---:|
| Menthol | 3.33 | $2.75 \pm 0.59$ | $3.96 \pm 0.85$ |
| **1c** | 4.68 | $3.50 \pm 1.66$ | $1.23 \pm 0.61$ |
| **1g** | 6.25 | $5.85 \pm 4.15$ | $2.99 \pm 2.12$ |

There is a clear positive correlation between the cLogP value and the $P_{app}$. Compound **1g**, with the highest cLogP value (6.25), exhibits significantly higher apparent permeability ($P_{app} = 5.85 \times 10^{-4}$ cm/s) compared to menthol and **1c**, which have cLogP values of 3.33 and 4.68, respectively. The higher lipophilicity of this derivative may play a crucial role in facilitating its transport through the biological membrane, as indicated by its enhanced $P_{app}$ compared to the derivative **1c**. This relationship suggests that lipophilicity plays a determining role in the compounds' ability to cross biological membranes, which aligns with theoretical knowledge. The comprehensive results obtained from the experimentation are in full agreement with the earlier predictions made using in silico techniques, where the O-carbonates of menthol (**1c** and **1g**) exhibited the highest $P_{app}$ values.

From the conducted assay, it can also be established that the three compounds considered in this study may be classified as high-permeability compounds. The rationale for this classification is based on their $P_{app}$ values, which were found to exceed that of metoprolol ($1.6 \times 10^{-5}$ cm/s). Metoprolol was intentionally chosen as a reference due to its established use by Delrivo et al. (2018) [32] in evaluating the boundary between high- and low-permeability compounds using the same permeability protocol with the same BAM.

Furthermore, a higher Jss value indicates a higher speed of flux of the drugs across the barrier, and, although lipophilicity (cLogP) may increase $P_{app}$, it does not necessarily lead to a higher drug flow rate through the barrier. In this case, menthol shows a Jss of $3.96 \times 10^{-5}$ mg/s, while **1c** has a lower value of $1.23 \times 10^{-5}$ mg/s and **1g** has a value of $2.99 \times 10^{-5}$ mg/s. Compound **1c** represents an interesting exception to this trend. Despite having an intermediate cLogP value, it displays a lower Jss compared to Menthol and **1g**, indicating slower flux permeation through the barrier. This discrepancy could be related to other specific factors of the compound, such as its molecular size or solubility.

In summary, the results suggest that lipophilicity plays a relevant role in compound permeability, but it is not the sole factor influencing permeation flux. These findings highlight the complexity of permeation processes and underscore the importance of considering multiple factors when designing and developing pharmacological compounds with the optimal capability to cross membranes and reach their therapeutic targets.

## 3. Discussion

Previous studies carried out in our research group showed that the menthol derivatives designed proved to be potent antiparasitic candidates, with moderate activity against total forms of *P. falciparum* and high activity against intracellular amastigotes of *T. cruzi* and *L. braziliensis* [23]. To start the preformulation studies of the menthol derivatives synthesized, in this work, we analyzed their chemical stability and permeation behavior. From

the results of this study, the successful development and validation of a GC method for the accurate determination and quantification of menthol and its prodrugs at various pH levels were efficiently executed. The results obtained from the chemical stability determined in aqueous solutions at different pH values showed that the prodrugs of menthol were degraded to the parent compound and not to other degradation products, which led to a simplification of the subsequent assays.

The carbonates were found to be more stable at a pH of 5.8. These profiles are in agreement with the stability data reported by other authors. For example, N'Da et al. developed Zidovudine carbonate prodrugs with the aim of improving the drug's pharmacokinetic properties and decreasing its side effects. They found that the chemical stability of each carbonate was significantly greater at a pH of 5.0 compared to a pH of 7.4 [31]. It is noteworthy that carbonate prodrugs hydrolyze more rapidly than carbamate and ester prodrugs [33]. The same behavior was observed when Dittert et al. studied the chemical stability of carbamate and carbonate compounds in different aqueous media [30].

In silico prediction of permeability showed that both, the starting compounds and their prodrugs, would passively permeate lipid membranes as they are hydrophobic molecules and, therefore, their distribution will be more advantageous within the hydrophobic core of the lipid bilayer rather than at the lipid–water interface [34,35]. For all compounds, it was predicted that they perform "flip-flop" movements in which, depending on the environment, they establish the energetically most optimized position to cross lipid membranes.

In terms of in vitro permeability studies, the BAM used was designed to closely mimic features of eukaryotic cellular membranes, including macrophage cells. We hypothesized that permeability assays can give a framework for comprehending the biological results obtained. These small molecules can cross plasma membranes into cells via simple permeation, accumulating first in the hydrophobic regions of the lipid bilayer due to hydrophobic interactions. Here, menthol derivatives have been formulated to enhance the cellular absorption of menthol by increasing their lipophilicity and promoting the cellular uptake by increasing the passage through membranes by simple diffusion, thus allowing more efficient accumulation inside the macrophages where the parasites evaluated were hosted [36].

## 4. Materials and Methods

### 4.1. General Procedure for the Synthesis of Carbonates of Menthol

The corresponding prodrugs, obtained by association of menthol with different aliphatic alcohols, were prepared as previously reported [23].

### 4.2. Stability Studies

#### 4.2.1. GC Conditions

The following equipment were used to study GC conditions: an Agilent 7890A chromatograph ZB-WAX column (Agilent, Santa Clara, CA, USA) (60 m length × 0.250 mm width × 0.25 μm film) and a flame ionizer. The set temperature limit was 20–250 °C, and Helium was used as the carrier gas with a flow rate of 2.00 mL/min.

#### 4.2.2. Validation of GC Method

Linearity. Ten concentrations equally distributed between 0.0001 and 0.1 M of menthol and its derivatives were selected, and the corresponding calibration curves were obtained. The data for each compound were obtained in triplicate.

Specificity. To calculate the specificity of the selected analytical method, the retention times of menthol and its derivatives and the absence of analyte interferences at different pH values were identified.

Limit of detection (LOD) and limit of quantification (LOQ). To determine LOD, the lowest analyte amount in the sample that could be detected but could not be quantified with an exact value was analyzed. In addition, the LOQ was calculated as the lowest amount of analyte in a sample that could be quantitatively determined with adequate precision and accuracy. Both parameters were calculated from the three lowest concentrations of

the calibration curve of each compound, and the data for each compound were obtained six times.

Recovery. Three solutions of known amounts of each compound corresponding to low, medium, and high concentrations were analyzed. The percent recovery was determined where the expected area had been calculated from the calibration curve.

Precision. Six injections of two concentrations (one high and one low) were used to calculate precision through relative standard deviation (RSD).

### 4.2.3. Chemical Stability in Aqueous Buffers

Stability studies of menthol and its derivatives were performed at pH = 1.2, pH = 5.8, and pH = 7.4 at 37 °C. The selected pH values of 1.2, 5.8, and 7.4, along with the physiological temperature of 37 °C, aimed to simulate the conditions encountered in the human body.

In specific, the choice of pH values (1.2, 5.8, and 7.4) and physiological temperature (37 °C) was made with the purpose of mimicking the conditions found in the human body. These specific pH values were selected to represent the acidic environment of the stomach (pH = 1.2), the pH of the upper gastrointestinal tract (pH = 5.8), and the pH of the blood and interstitial fluids (pH = 7.4). By subjecting the samples to these simulated physiological conditions, we aimed to gain insights into the behavior and stability of the compounds under realistic in vivo scenarios.

A 20 mL buffer was equilibrated at 37 °C for 10 min before introducing each compound diluted in dichloromethane to give an initial concentration of $5 \times 10^{-5}$ mol/mL, which was then mixed using a vortexer for 30 s. Samples of 1 mL were extracted at appropriate time intervals. For quantification of menthol prodrugs in GC, 1 mL of dichloromethane was added to each sample, and then the organic phase was extracted and quantified. By applying linear regression to the natural logarithm of the concentration vs. time plot, the pseudo-first order rate constant of the prodrugs was determined. The data for each compound were obtained in triplicate.

The half-life time ($t_{1/2}$) for a first-order reaction was calculated according to Equation (1):

$$t_{1/2} = \frac{2}{k} \tag{1}$$

### *4.3. Permeability Studies*
### 4.3.1. In Silico Assays

The open source software UCSF Chimera 1.15 was used to build and optimize the three-dimensional structures of the compounds, and the PyMol program was used for their visualization. Additionally, we predicted passive membrane permeation across a DOPC bilayer using the PerMM online server [37] based on the inhomogeneous solubility diffusion model. For in silico permeability analysis, the PerMM server was used at T = 298 K, pH = 5.8. The choice of a pH value of 5.8 for the in silico study was made to anticipate the mode of passage through the BAM under these conditions. The decision not to predict permeability at pH 7.4 was based on the understanding that this pH corresponds to the receptor compartment, where the sample is already assumed to be permeated.

### 4.3.2. In Vitro Assays

Permeability analyses were conducted using a home-built side-by-side diffusion cell composed of two half-cells connected to a water bath at 37 °C, equipped with a water jacket, separated by a BAM, and constructed by impregnating Lipoid 75 in 10% (*w/v*) n-octanol into a cellulose ester support following the procedure described in Delrivo et al. (2018) [32]. The BAM was put in between the donor and receptor compartments after reaching equilibrium in the corresponding buffer solution. Transport studies were performed with the donor compartment filled with 4 mL of 0.287, 0.072, and 0.102 mg/mL of drug solution (menthol or one of its two prodrugs, menthol–propanol (**1c**) or menthol–hexanol (**1g**), respectively) in 4.0 mL of a buffer solution at a pH value of 5.8 and the receptor

compartment being filled with phosphate-buffered saline (PBS) at a pH environment of 7.4. Both compartments had 3% (*v/v*) of DMSO to ensure drug solubility and sink conditions. Both compartments were also constantly stirred. Then, 1 mL of the receptor fluid samples were taken at determined times and the same volume was replaced with PBS at a pH value of 7.4 with 3% (*v/v*) of DMSO solution at 37 °C. The drug concentrations were determined by GC analyses. Each experiment was conducted in triplicate.

The drug's permeation flux (Jss, mg/cm$^2$·s) and apparent permeability (P$_{app}$, cm/s) were subsequently determined by analyzing the slope of the linear segment in the cumulative drug permeation profiles across the BAM over time.

The drug flux was determined from Fick's law of diffusion according to the Equation (2):

$$Jss = \frac{dQr}{A}dt \qquad (2)$$

In this equation, Jss represents the steady-state flux, dQr denotes the change in drug quantity traversing the BAM into the receptor compartment, A signifies the effective diffusion area in cm$^2$ (1.44 cm$^2$), and dt signifies the change in time. The P$_{app}$ was determined using Equation (3):

$$Papp = \frac{Jss}{Co} \qquad (3)$$

where Co is the initial drug concentration in donor solutions.

## 5. Conclusions

Monoterpenes, as natural compounds, display a broad spectrum of biological effects, including antimicrobial, anti-inflammatory, anticancer, and antiparasitic effects, as we have previously demonstrated. Their lipophilic nature facilitates efficient traversal across biological membranes, driven by hydrophobic interactions with the lipid bilayer. Understanding these mechanisms is pivotal for harnessing the pharmacological actions and therapeutic potential of monoterpenes, paving the way for developing innovative drugs and therapeutic strategies.

The utilization of a carbonate linkage, which connects the hydroxyl (OH) position of menthol to diverse aliphatic alcohols, represents an intriguing and promising approach aimed at enhancing both the safety and effectiveness of this monoterpene compound. As potential prodrugs, these compounds hold great potential as viable therapeutic agents for the treatment of infectious diseases.

**Author Contributions:** Conceptualization, C.M.C., G.E.G. and S.R.; methodology, C.M.C., R.O. and N.U.; software, C.M.C.; validation, C.M.C., R.O. and S.R.; formal analysis, C.M.C., R.O., G.E.G. and S.R.; resources, S.R.; writing—original draft preparation, C.M.C., R.O., G.E.G. and S.R.; writing—review and editing, C.M.C. and S.R.; visualization, C.M.C. and S.R.; supervision, C.M.C. and S.R.; project administration, S.R. All authors have read and agreed to the published version of the manuscript.

**Funding:** This study was supported by Universidad Nacional de Villa María and Consejo Nacional de Investigaciones Científicas y Técnicas de Argentina (RES2021.024, RES2021.11, Res2021.384 and RES2022.358. 210.000 ARS).

**Institutional Review Board Statement:** Not applicable.

**Informed Consent Statement:** Not applicable.

**Data Availability Statement:** Data are contained within the article.

**Acknowledgments:** The authors acknowledge their respective universities and institutions for their support. In addition, we thank Mariana Bonaterra (IMITAB-UNVM) for their cooperation in GC analysis.

**Conflicts of Interest:** The authors declare no conflict of interest.

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
