# Peer review of "Preformulation Studies of Novel Menthol Prodrugs with Antiparasitic Activity: Chemical Stability, In Silico, and In Vitro Permeability Assays"

_ddc, doi:10.3390/ddc2030038_

Round 1

Reviewer 1 Report

The authors should include why they did not perform the in silico assay using pH 7.4 too.

Author Response

Reviewer 1:

  1. The authors should include why they did not perform the in silico assay using pH 7.4 too.

The permeability prediction was conducted to gather preliminary insights before the in vitro assay involving the BAM and to anticipate a potential mechanism of membrane passage. The choice of pH 5.8 for in silico study was made to anticipate the mode of passage through the BAM under these conditions. The decision not to predict permeability at pH 7.4 was based on the understanding that this pH corresponds to the receptor compartment, where the sample is already assumed to be permeated. This description is highlighted in section 4.4.1. of the manuscript.

Reviewer 2 Report

Overall, the paper discussed by Clemente et al is a nice but short interesting piece of work. I think that the work is interesting fo rth ereaders of the journal howerver, there are some points that need clarification before proceeding with publication. 

1. The methodology should be more detialed specially to descirbed how the values are obtained from the free software. Do you have more references that corroborate the usefulness of this software? 

2. The seocnd point to consider is what happen if the molecule is encapsualted within a carrier suhc as trnasferosomes. Drugs cna be encapsualted within carriers to enhance their permeability across the skin. How do you envisage that this strategy will affect the permeability of the cited ocmpounds? I think the discussion has to be expanded. See papers: "Transferosomes as nanocarriers for drugs across the skin: Quality by design from lab to industrial scale" or "Topical buparvaquone nano-enabled hydrogels for cutaneous leishmaniasis"

3. Have you considered ot use other type of membranes such as Strat- membranes or human or pig skin?

4. English grammar has to be revised. 

English needs to be revised. 

Author Response

Reviewer 2:

Overall, the paper discussed by Clemente et al is a nice but short interesting piece of work. I think that the work is interesting for the readers of the journal howerver, there are some points that need clarification before proceeding with publication. 

We thank the reviewer for his/her positive comment on the manuscript. We have carefully read all reviewer’s suggestions and revised the manuscript accordingly. 

  1. The methodology should be more detailed specially to describe how the values are obtained from the free software. Do you have more references that corroborate the usefulness of this software? 

The free software PerMM is a predictive method that estimates permeation coefficient (at specified pH) of neutral and ionizable molecules and allows visualization of permeant movement through the DOPC bilayer. PerMM method combines the heterogeneous solubility-diffusion theory and the anisotropic solvent model of the lipid bilayer characterized by transbilayer profiles of dielectric and hydrogen-bonding capacity parameters. Several studies have compared the permeability coefficients calculated by this statistically based method showing that PerMM performs better than other known programs such as pkCSM and admetSAR. These reports are attached below:   

-Frallicciardi, J., Melcr, J., Siginou, P., Marrink, S. J., & Poolman, B. (2022). Membrane thickness, lipid phase and sterol type are determining factors in the permeability of membranes to small solutes. Nature Communications, 13(1), 1605.

-Róg, T., Girych, M., & Bunker, A. (2021). Mechanistic understanding from molecular dynamics in pharmaceutical research 2: lipid membrane in drug design. Pharmaceuticals, 14(10), 1062.

-Lomize, A. L., & Pogozheva, I. D. (2019). Physics-based method for modeling passive membrane permeability and translocation pathways of bioactive molecules. Journal of chemical information and modeling, 59(7), 3198-3213.

  1. The second point to consider is what happens if the molecule is encapsulated within a carrier such as transfersomes. Drugs can be encapsulated within carriers to enhance their permeability across the skin. How do you envisage that this strategy will affect the permeability of the cited compounds? I think the discussion has to be expanded. See papers: "Transferosomes as nanocarriers for drugs across the skin: Quality by design from lab to industrial scale" or "Topical buparvaquone nano-enabled hydrogels for cutaneous leishmaniasis"

The mentioned studies offer interesting perspectives as potential carriers of these prodrugs, and we thank them for sharing their findings. However, it is necessary to point out that in our in vitro permeability investigations, the Biomimetic Artificial Membrane (BAM) employed was meticulously designed to closely emulate the attributes of eukaryotic cell membranes, including those inherent to macrophage cells. Our working hypothesis postulated that these permeability assays could provide a conceptual framework for understanding the biological results obtained.

Our observations revealed that these small molecules could cross cellular plasma membranes by direct permeation, initially preferentially accumulating in the hydrophobic domains of the lipid bilayer, driven by hydrophobic interactions.

In this context, we designed menthol derivatives with the main objective of increasing the cellular uptake of menthol. This enhancement was achieved by increasing its lipophilicity and facilitating cellular entry by increasing its transmembrane passage through simple diffusion mechanisms. Consequently, this strategy facilitated a more efficient intracellular accumulation of menthol derivatives in macrophages, where the target parasites were predominantly located. Previous investigations by our research group had contemplated the possibility of oral administration of these prodrugs, but also an interesting route of exploration is their possible topical application.

  1. Have you considered ot use other type of membranes such as Strat- membranes or human or pig skin?

We appreciate your suggestions, and they will be taken into account in future studies. It is important to highlight that due to the potential of these new menthol prodrugs as antiparasitic candidates, one of the main objectives of the work was to determine the stability of compounds 1c and 1g (Figure 1) in aqueous solutions at different pH values of biological relevance, as well as the permeability of the prodrugs and their parent compound through in silico and in vitro assays using a biomimetic artificial membrane (BAM) that simulates the behavior of eukaryotic cell membranes.

To study permeability in the early stages of drug design and development, PAMPA (Parallel Artificial Membrane Permeability Assay) studies are normally used, which allow for simple physicochemical determination of the permeability of compounds with low cost and high throughput. However, the disadvantage of this technique is the small volume of the donor and acceptor compartments within the plates, which does not maintain saturation conditions during permeation experiments, nor does it generate a stirring condition. What we propose is an in vitro permeation method that was previously developed and validated in our laboratory, involving the use of Franz diffusion cells with biomimetic artificial membranes. This method is a simple, fast, and useful technique for predicting permeation coefficients with good reproducibility and low cost. Additionally, it has the advantage of being able to control temperature, allowing the donor and acceptor cells to be kept under continuous agitation. In comparison to PAMPA, the donor and acceptor compartments have a larger volume, preserving saturation conditions during the permeation experiment. For these reasons, at this early stage of analysis, this technique contributes to replacing and reducing the use of research animals.

  1. English grammar has to be revised. 

English grammar has already been reviewed by a native speaker.

Reviewer 3 Report

The manuscript describes the preformulation studies of novel menthol prodrugs with antiparasitic activity and it seems to be well designed. However, there are some aspects that should be improved:

Lines 33-40 and 58-70: is missing some references to support these ideas contained in the paragraph.

Lines 182-185 and 187-190: the information is almost repeated. Regarding the information that metoprolol has a higher value this higher values it was observed in the study of reference, for the most accurate validation and comparison of the Papp also the metoprolol should be evaluated in your study, and the inclusion of a low permeability compound was also essential (not only a high permeability control). 

Author Response

Reviewer 3:

The manuscript describes the preformulation studies of novel menthol prodrugs with antiparasitic activity and it seems to be well designed. However, there are some aspects that should be improved.

We thank the reviewer for his/her positive comment on the manuscript. We have carefully read all reviewer’s suggestions and revised the manuscript accordingly. These are highlighted in yellow in the corresponding section.

1. Lines 33-40 and 58-70: is missing some references to support these ideas contained in the paragraph.

The references supporting the ideas presented in the introduction have been cited and highlighted in yellow.

2. Lines 182-185 and 187-190: the information is almost repeated. Regarding the information that metoprolol has a higher value this higher values it was observed in the study of reference, for the most accurate validation and comparison of the Papp also the metoprolol should be evaluated in your study, and the inclusion of a low permeability compound was also essential (not only a high permeability control).

Thank you for your feedback. We have already addressed the issue of repeated information in lines 182-185 and 187-190, and those paragraphs have been revised accordingly. Regarding the suggestion to evaluate metoprolol in our study for a more accurate validation and comparison of the Papp, we would like to clarify that the article cited for metoprolol was conducted within our research group, and the methods used were validated under identical conditions. Additionally, another reason for not including a high and low permeability compound as a control in our study was the lack of availability of BAM equipment in our laboratory.

Round 2

Reviewer 2 Report

Authors have addressed the comments accordingly so the manuscrito is ready for publication.

Ok minor errors are encountered.